# Fault Diagnosis of Railway Point Machines Using the Locally Connected Autoencoder

**Zhen Li** [1,2,3,*], **Zhuo Yin** [2,3] , **Tao Tang** [1] **and Chunhai Gao** [2,3]

1   School of Electronic and Information Engineering, Beijing Jiaotong University, Beijing 10044, China;
    ttang@bjtu.edu.cn
2   National Engineering Laboratory for Urban Rail Transit Communication and Operation Control,
    Beijing 100044, China; michaelyin1994@emails.bjut.edu.cn (Z.Y.); chunhai.gao@bj-tct.com (C.G.)
3   Traffic Control Technology Co., Ltd., Beijing 100070, China
*   Correspondence: 12111041@bjtu.edu.cn; Tel.: +86-10-5282-4695

**Abstract:** Data-driven fault diagnosis is considered a modern technique in Industry 4.0. In the area of urban rail transit, researchers focus on the fault diagnosis of railway point machines as failures of the point machine may cause serious accidents, such as the derailment of a train, leading to significant personnel and property loss. This paper presents a novel data-driven fault diagnosis scheme for railway point machines using current signals. Different from any handcrafted feature extraction approach, the proposed scheme employs a locally connected autoencoder to automatically capture high-order features. To enhance the temporal characteristic, the current signals are segmented and blended into some subsequences. These subsequences are then fed to the proposed autoencoder. With the help of a weighting strategy, the seized features are weight averaged into a final representation. At last, different from the existing classification methods, we employ the local outlier factor algorithm to solve the fault diagnosis problem without any training steps, as the accurate data labels that indicate a healthy or unhealthy state are difficult to acquire. To verify the effectiveness of the proposed fault diagnosis scheme, a fault dataset termed "Cu-3300" is created by collecting 3300 in-field current signals. Using Cu-3300, we perform comprehensive analysis to demonstrate that the proposed scheme outperforms the existing methods. We have made the dataset Cu-3300 and the code file freely accessible as open source files. To the best of our knowledge, the dataset Cu-3300 is the first open source dataset in the area of railway point machines and our conducted research is the first to investigate the use of autoencoders for fault diagnosis of point machines.

**Keywords:** railway point machines; urban rail transit systems; fault diagnosis; autoencoder

## 1. Introduction

With the tremendous deployment of different kinds of sensors and actuators, the Internet of Things (IoT) emerges as an advanced method to connect devices and collect the status data [1]. Aided by the use of a large amount of operation data, the data-driven fault diagnosis is considered as a modern technique in Industry 4.0 and has become a research hotspot in recent years [2–5]. In the area of urban rail transit, the significant increase of the line mileage and the passenger throughput leads to a high capacity utilization of the existing infrastructure [6]. This kind of situation may cause more equipment failures and service disruptions, resulting in a great impact on traffic safety, property, and customer satisfaction. Among these infrastructure failures in urban rail transit systems, the vast majority of them are triggered by railway point machines [7]. The point machine is a critical component that is used to safely switch the train direction. To meet the demand of fault-tolerant operation of

railway transportation and reduce the maintenance costs of metro operators, advanced fault diagnoses schemes of the point machines are of utmost need.

The literature includes a wide range of methods for condition monitoring-based failure diagnosis of point machines, including sound analysis [8], gap measurement [9,10], and electric current analysis [11–13]. Among these methods, electric current analysis is considered a straightforward and effective approach for failure diagnosis of the point machine, as the point machine is directly actuated by an electric motor [14]. The support vector machine is a representative method to solve the task [15]. The support vector classification method usually has two key steps: feature extraction and fault recognition. The feature extraction process, however, is typically manually designed—as the classifier is incapable of capturing information from raw signals—and typically much of the main effort involves incorporating expert knowledge into the system, which can be influenced by personal attitudes and not fully objective, like artificial intelligence. The problem of how to automatically extract features needs to be well addressed. In addition, one core characteristic of our data is that each current signal is a time series of correlated count data. The temporal characteristic should be taken into consideration to improve the accuracy of fault diagnosis of point machines [16].

Bearing these challenges in mind, in this paper, we propose a novel fault diagnosis scheme for railway point machines. First, a locally connected autoencoder is employed to obtain high-order feature representations. Thereby, instead of the labor-intensive hand-engineering feature extraction approach, effective features can automatically be captured from the raw electrical data. Different from other autoencoder-based methods, in the proposed autoencoder, "locally" means that the current signal is equally segmented into some subsequences. Then, these subsequences are fed sequentially to the input layer of the autoencoder. To enhance the temporal characteristic, inspired by the word2vec method [17], adjacent subsequences are blended to obtain the features of the current subsequence. Second, after the local features are extracted using the locally connected autoencoder, a weighting strategy is proposed to weight average the features into their final representations. The reason for using this strategy is that the electric signals collected from the point machines may contain some noninformative information, which causes the unexpected redundancy and the high disturbance of the local features. By employing a weighting function, useful and robust features will be enhanced, whereas the less informative features will be reduced. Finally, different from the existing methods, which input the features into the softmax classifier for classification, we use the Local Outlier Factor (LOF) algorithm for outlier detection. The function of the softmax classifier is to give the probabilities for each class label [18]. The LOF algorithm does not require prelabeled samples for its learning process, and therefore is very suitable for fault diagnosis of point machines, because the samples gathered in practice related to such machines typically are in lack of labels. Further, the fault datasets integrated by the researchers in general also lack completeness, as not all types of possible unhealthy states occur within the datasets.

To verify the effectiveness of the proposed fault diagnosis scheme, we collected 3300 in-field current signals to be used for fault analysis by various algorithms. The signals are generated from point machines that are installed and currently being used in operation lines. By aid of this dataset, we evaluate the prediction performance of the proposed algorithm by comparing its prediction skills with respect to existing methods, such as the feature-based LOF algorithm, the stacked denoising autoencoder, the sparse denoising autoencoder, and the gated recurrent unit-based sequence to sequence autoencoder. The results show that the proposed scheme not only has the ability to automatically capture most effective features from the raw current signals, but also achieves superior diagnosis accuracy compared to the existing methods.

The fault dataset, which we have termed Cu-3300, as well as the code file of the proposed scheme have been made available online at https://github.com/MichaelYin1994/SignalRepresentationAnalysis. To the best of our knowledge, Cu-3300 is the first open source dataset in the area of railway point machines and our fault analysis is the first to investigate the use of autoencoders for fault diagnosis of point machines.

The rest of the paper is organized as follows. Section 2 discusses the related work. The proposed fault diagnosis scheme is detailed in Section 3. Section 4 verifies the superior performance of the proposed scheme. Section 5 concludes the paper.

## 2. Related Works

Fault diagnosis of point machines has been widely addressed in urban rail transit. Traditionally, condition monitoring systems are applied to rail transit applications to ensure driving safety. In these systems, a simple threshold is designed to detect abnormal behavior, resulting in poor performance of the detection accuracy. To improve the condition monitoring system, much effort has been targeted towards capturing the fault patterns. Shaw [19] investigated various point machines and explored core measurements for fault diagnosis. In this work, the electrical signals are considered to be the foremost feature to study to infer the health condition of the point machine. Vileiniskis et al. [13] presented an early detection method in the measurement of the current signals. Using the similarity measure of edit distance, the one-class support vector machine is employed to seize the changes. Using the electric current as a parameter, Asada et al. [7] proposed a wavelet transform-based feature extraction scheme by which an accurate health prediction was obtained. Aided by in-field current data, the authors of [14] proposed a classification method to detect the replacement conditions. Sa et al. [11] focused on the aging effect of the point machine. This task is regarded as a binary-class classification problem. Similarly, the support vector data description was applied. Different from the current data-based approach, the authors of [8] put forward a data mining scheme that employs audio data of the point machine. The scheme is capable of extracting mel-frequency cepstrum coefficients to reduced feature dimensions. The fault detection accuracy is reported to exceed 94.1%. The point machines' gap was investigated in [10]. The proposed edge detection algorithm has the advantage of overcoming the local intensity variation in gap images, which can effectively show the health condition of the point machines.

All the research discussed above are classification approaches, where the training process requires prelabeled samples. These researches mentioned above can be categorized as the classification approach, where the training process is necessary with prelabeled samples. However, the lack of available labeled data necessary for the training of the models used for anomaly detection techniques is usually a major obstacle to apply such models. Kim et al. [12] pioneered a diagnosis method using dynamic time warping to manage the variation of the current signals without training steps. Inspired by the authors of [12], we attempt to solve the fault detection problem in a non-training way. In this paper, the LOF algorithm is employed for fault detection.

Autoencoders have attracted attention in the field of fault diagnosis as replacements for any handcrafted features and expert knowledge incorporated in the traditional fault analysis approaches. Thirukovalluru et al. [20] presented a review on handcrafted based feature extraction algorithms and autoencoders. In this paper, a fault analysis was performed based on five different fault datasets and it was demonstrated that the autoencoder methods performed best. A two-layer encoder was proposed in [21], which was regularized by the contractive regularization and validated on artificially constructed datasets. Using bearing and gearbox fault datasets, the authors of [4] showed that the autoencoder yields better classification performances than the support vector machine based methods. The authors of [22] used a parse autoencoder to obtain useful representations of different sensor signals. The obtained representations were fed to train a deep belief network for fault classification. To address for unobserved working conditions, the authors of [23] developed a cross-domain stacked denoising autoencoder. The maximum mean discrepancy was calculated to achieve effective marginal distribution adaptation. Wu et al. [24] proposed a multilevel denoising autoencoder approach. By accounting for the effects of hyperparameters, their approach outperformed traditional approaches, such as Principal Component Analysis (PCA) and Fisher Discriminant Analysis. However, this effective automatic feature learning technique has not been introduced in fault diagnosis of the railway point machine. Therefore in this paper, adapting the inherent characteristics of the point machine, we employ

the locally connected autoencoder to locally capture any informative feature. Then, a weighting strategy is designed to integrate the extracted features into useful representations.

## 3. Methodology

### 3.1. Current Signals of Railway Point Machines

A railway point machine consists of a motor, a transmission clutch, gearing, and a retention clutch with throw bar. The function of the point machine is to actuate the switch rail from one stock rail to the opposite position. The failure of the point machine may cause serious accidents, such as the derailment of the train with all its consequences. The health condition of the point machine is closely related to the process of the switch operation. Thus, it is particularly useful to monitor the switch operation process. The current sensors that are widely deployed in field could directly collect the electrical signals, which provides sufficient samples for the electric current analysis to solve the fault diagnosis problem. The operation of the point machine can be divided into the following four phases.

1. The start-up phase: An activating signal transmitted from the control center is received by the point machine. The asynchronous short-circuit three-phase motor starts up. One can find a current surge in this phase.
2. The unlocking phase: At the beginning of the throwing process, the switch rail is unlocked. The current signal is observed to be gradually declined.
3. The switch phase: The actual throwing process begins. The switch rail smoothly moves on the shifting plate.
4. The locking phase: At the end of the throwing movement, when the switch rail arrives at the end position, one keep-and-detect slide engages with the end position notch and locks the retention clutch. Simultaneously, a signal is sent from the point machine to report the successful operation.

A representative current shape which shows the four operating phases is shown in Figure 1.

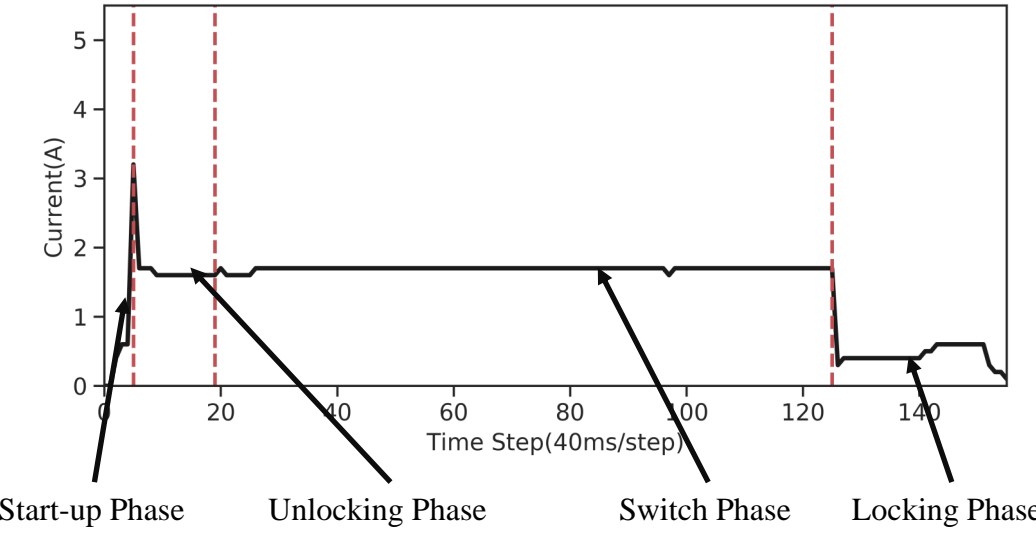

**Figure 1.** Operating phases of the point machine. See the text for additional detail.

### 3.2. The Proposed Locally Connected Autoencoder

Figure 2 shows the overall structure of the proposed fault diagnosis system. First, the current signals are generated from the point machine and are captured by the current sensors. The signals are then transmitted by an LTE communication device to the cloud platform where the signals are stored. The proposed fault diagnosis algorithm is deployed on the cloud platform. The network structure of a typical autoencoder consists of an input layer, a hidden layer, and an output layer [25]. As a small

alteration of the typical neural network, the output target of the autoencoder is set to be the input data, that is, the output should be as close to identical as possible to reconstruct the input data. Therefore, the autoencoder can be regarded as an unsupervised learning process since it does not require labeled data. The input layer and the hidden layer constitute the encoder part. The hidden layer generally has fewer neurons, where the high-dimensional raw signals are mapped onto the low-dimensional feature space. Thereby, a kind of the high-order feature representation is seized. Conversely, the decoder part consists of a hidden layer and an output layer, which reconstructs the input signal from the extracted features.

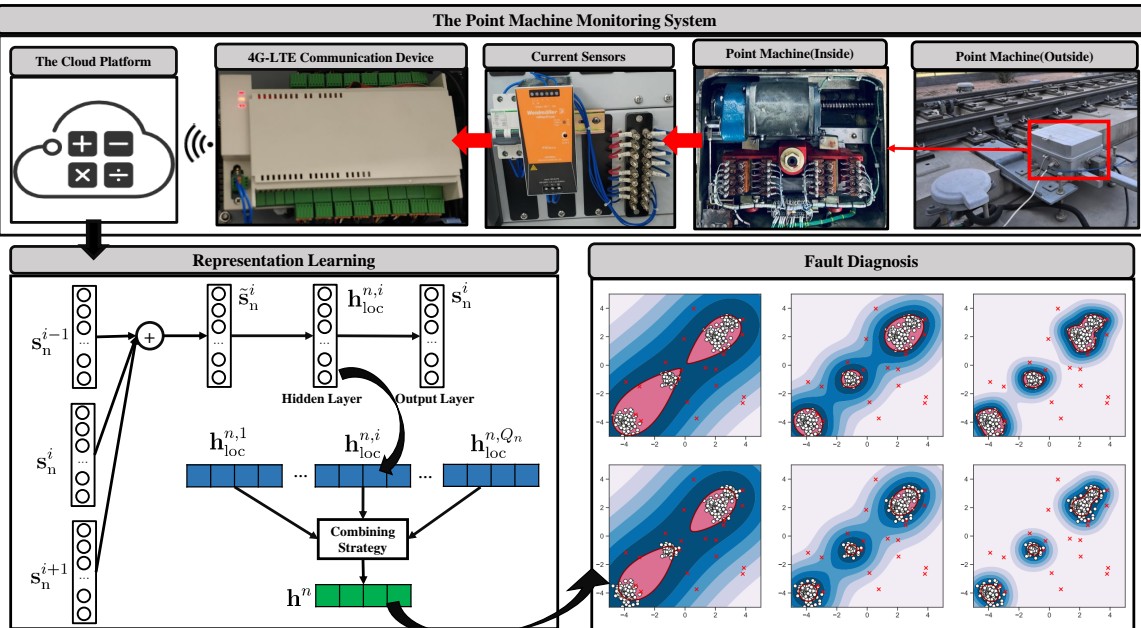

**Figure 2.** The flow chart of the proposed fault diagnosis scheme of the point machine.

Let $\mathbb{S} = \{\mathbf{s}_1, \mathbf{s}_2, \cdots, \mathbf{s}_N\}$ denote the set of current signals of the point machines. For a certain signal $\mathbf{s}_i$, we define

$$\mathbf{h}_i = f(\mathbf{s}_i; \Theta),$$

where $f(\cdot)$ represents the encoding function. $\mathbf{h}_i$ is the encoder vector that represents the feature of the input signals and $\Theta$ is the set of the encoder parameters.

Similarly, for the decoder part, we define

$$\mathbf{y}_i = g(\mathbf{h}_i; \Theta'),$$

where $\Theta'$ is defined as the set of the decoder parameters. $\mathbf{y}_i$ is the network output and $\mathbb{Y} = \{\mathbf{y}_1, \mathbf{y}_2, \cdots, \mathbf{y}_N\}$ denotes the output set. $g(\cdot)$ represents the reconstruction function.

Generally, an affine function is employed in $f(\cdot)$ and $g(\cdot)$, that is,

$$f(\mathbf{s}_i; \Theta) = \sigma_f(\mathbf{W}\mathbf{s}_i + \mathbf{b}),$$

$$g(\mathbf{h}_i; \Theta') = \sigma_g(\mathbf{W}'\mathbf{h}_i + \mathbf{b}'),$$

where $\Theta = \{\mathbf{W}, \mathbf{b}\}$ and $\Theta' = \{\mathbf{W}', \mathbf{b}'\}$ and $\mathbf{W}$ and $\mathbf{b}$ are, respectively, the weight matrix and the bias vector of the encoder part, and $\mathbf{W}'$ and $\mathbf{b}'$ are, respectively, the weight matrix and the bias vector of the decoder part. Further, $\sigma_f(\cdot)$ and $\sigma_g(\cdot)$ represent the activation functions used to seize nonlinear relationships.

To describe the average differences (also termed as the reconstruction errors) between the outputs of the decoder part and the actual values, a loss function is defined and calculated as

$$\min_{\mathbf{W},\mathbf{W}',\mathbf{b},\mathbf{b}'} \frac{1}{N} \sum_{i=1}^{N} L(\mathbf{s}_i, \sigma_g(\sigma_f(\mathbf{s}_i))), \tag{1}$$

where

$$L(\mathbf{x}, \mathbf{x}') = \|\mathbf{x} - \mathbf{x}'\|_2^2.$$

Note that in the above-mentioned autoencoder, each training sample is directly fed to the input layer. Thus, the extracted features are global descriptions that may lack the representation of local properties. To enhance the local properties of the fault signals, we divide the raw signals into segments for feature extraction. For each current signal $\mathbf{s}_n$ in dataset $\mathbb{S}$, $\mathbf{s}_n \in \mathfrak{R}^{|\mathbf{s}_n| \times 1}$ denotes the $n$th signal with $|\mathbf{s}_n|$ data points. The signal $\mathbf{s}_n$ is equally segmented into $Q_n$ subsequences

$$\mathbf{s}_n = \{\mathbf{s}_n^1, \mathbf{s}_n^2, \cdots, \mathbf{s}_n^{Q_n}\}, Q_n = \lfloor |\mathbf{s}_n| / M_{\text{loc}} \rfloor,$$

where $M_{\text{loc}}$ denotes the window size, $\mathbf{s}_n^i \in \mathfrak{R}^{M_{\text{loc}} \times 1}$, and $i \in \{1, 2, \cdots, Q_n\}$. Note that the number of segments generated by $\mathbf{s}_n$ may not be the same, as different signals typically consist of a different number of points.

Different from the existing methods [21–24], to obtain richer feature representations, in this paper, we propose a new network structure—the locally connected autoencoder—to enhance the temporal characteristics by preserving the linear regularities among subsequences. Inspired by the word2vec method [17], we define

$$\tilde{\mathbf{s}}_n^i = \mathbf{s}_n^{i-1} \oplus \mathbf{s}_n^i \oplus \mathbf{s}_n^{i+1}, \tag{2}$$

where $\oplus$ represents vector addition.

Using Equation (2), we create a new dataset $\tilde{\mathbb{S}} = \{\tilde{\mathbf{s}}_1, \tilde{\mathbf{s}}_2, \cdots, \tilde{\mathbf{s}}_N\}$, where

$$\tilde{\mathbf{s}}_n = \{\tilde{\mathbf{s}}_n^1, \tilde{\mathbf{s}}_n^2, \cdots, \tilde{\mathbf{s}}_n^{Q_n}\}.$$

$\tilde{\mathbf{s}}_n^i$ is fed into the encoder and the locally connected encoder is used to extract the local feature from $\mathbf{s}_n^i$.

We randomly select the segments $\tilde{\mathbf{s}}_n^i \in \mathfrak{R}^{M_{\text{loc}} \times 1}$ from the dataset $\tilde{\mathbb{S}}$ to compose the local training set. Then, $\mathbf{W} \in \mathfrak{R}^{K_{\text{loc}} \times M_{\text{loc}}}$, $\mathbf{b} \in \mathfrak{R}^{K_{\text{loc}} \times 1}$, $\mathbf{W}' \in \mathfrak{R}^{M_{\text{loc}} \times K_{\text{loc}}}$, and $\mathbf{W}' \in \mathfrak{R}^{M_{\text{loc}} \times 1}$ are trained in the network, where $K_{\text{loc}}$ denotes the hidden size. In the encoder part, $\mathbf{W}$ and $\mathbf{b}$ are used to map $\tilde{\mathbf{s}}_n^i$ into the feature space, whereas $\mathbf{W}'$ and $\mathbf{b}'$ are employed to reconstruct $\tilde{\mathbf{s}}_n^i$. The minibatch gradient descent method is employed with $N_s$ samples. Equation (1) can be rewritten as

$$\min_{\mathbf{W},\mathbf{W}',\mathbf{b},\mathbf{b}'} \frac{1}{N_s} \sum_{n=1}^{N_s} \|\sigma_g[\mathbf{W}' \sigma_f(\mathbf{W} \tilde{\mathbf{s}}_n^i + \mathbf{b}) + \mathbf{b}'] - \mathbf{s}_n^i\|_2^2,$$

where $\sigma_f(\cdot)$ is defined as the hyperbolic tangent function, i.e., $tanh(\cdot)$, to capture the nonlinear relationship, whereas $\sigma_g(\cdot)$ is set as the linear function. The hyperbolic tangent function can be written as

$$\tanh(x) = \frac{e^x - e^{-x}}{e^x + e^{-x}}.$$

The backpropagation algorithm and minibatch gradient descent method are used to train and update all the parameters to reduce the reconstruction error, where the Adam optimizer is adopted. In addition, the early stopping mechanism is used to limit overfitting. The derivations of the training process are not covered in this paper due to space restrictions. One can refer to the work in [25] for a detailed description.

After the training process, the local feature $\mathbf{h}_{\text{loc}}^{n,i} \in \mathfrak{R}^{K_{\text{loc}} \times 1}$ of the segment $\tilde{\mathbf{s}}_n^i$ can be calculated as

$$\mathbf{h}_{\text{loc}}^{n,i} = \tanh[\mathbf{W}(\mathbf{s}_n^{i-1} \oplus \mathbf{s}_n^i \oplus \mathbf{s}_n^{i+1}) + \mathbf{b}].$$

The encoding result $\mathbf{h}_{\text{loc}}^{n,i}$ preserves useful information of the segment $\mathbf{s}_n^i$ and can be used to form the representation of the original signal $\mathbf{s}_n$.

### 3.3. The Proposed Weighting Strategy

In the autoencoder-based method $\mathbf{h}_{\text{loc}}^{n,i}$ is designed as the high-order feature with $K_{\text{loc}} < M_{\text{loc}}$, which is automatically extracted. In [15], the simple average strategy is used to combine the local features into a final representation. Although sometimes the average strategy is appropriate to use, often it will have a bad influence on the final signal representations of the point machine, as different subsequences contain typically very different amounts of information. For example, the subsequences related to the switch phase may contain less useful information then the other stages, as the current signal remains at an almost constant level. To obtain more effective final feature representations, we proposed a weighting strategy as depicted in Figure 3 and explained below.

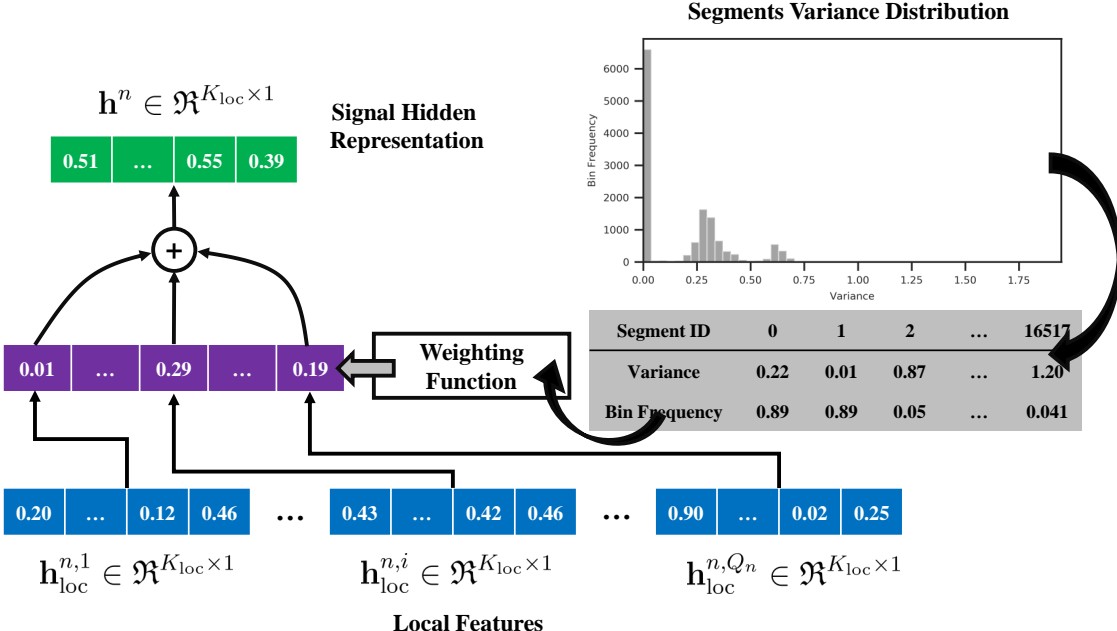

**Figure 3.** The proposed weighting strategy.

For a certain subsequence $\mathbf{s}_n^i = [s_n^{i,1}, s_n^{i,2}, \cdots, s_n^{i,M_{\text{loc}}}]$, the segment variance is calculated as

$$v_n^i = \frac{\sum_{j=1}^{M_{\text{loc}}} (s_n^{i,j} - \mu_n^i)^2}{M_{\text{loc}}},$$

where $\mu_n^i = \frac{\sum_{j=1}^{M_{\text{loc}}} s_n^{i,j}}{M_{\text{loc}}}$. Then, we create a variance dataset $\mathbb{S}_{\text{Var}}$ defined by $\mathbb{S}_{\text{Var}} = \{\mathbf{v}_1, \mathbf{v}_2, \cdots, \mathbf{v}_N\}$, where $\mathbf{v}_n = \{v_n^1, v_n^2, \cdots, v_n^{Q_n}\}$.

Then, the variance dataset $\mathbb{S}_{\text{Var}}$ is equally divided into $L$ bins, where $\mathbb{S}_{\text{Var}} = \{\mathbb{S}_{\text{Var}}^1, \mathbb{S}_{\text{Var}}^2, \cdots, \mathbb{S}_{\text{Var}}^L\}$. We define a parameter $\rho$ to describe the bin frequency, where $\rho_l$ denotes the number of variances located in $\mathbb{S}_{\text{Var}}^l$.

A weighting function is designed as

$$\omega(\mathbf{s}_n^i) = \frac{2}{1 + e^{\alpha \cdot \rho_l}}, \text{ if } v_n^i \in \mathbb{S}_{\text{Var}}^l.$$

Here, $\alpha$ is defined as the tuning parameter. Then, the final hidden representation of a certain current signal $\mathbf{s}_n$ is calculated as

$$\mathbf{h}^n = \frac{\sum_{i=1}^{Q_n} \omega(\mathbf{s}_n^i)\mathbf{h}_{\mathrm{loc}}^{n,i}}{Q_n}.$$

By aid of this analysis, it is clear that the high-order representation $\mathbf{h}^n$ is captured from $\mathbf{s}_n$ by the locally connected autoencoder and the weighting strategy.

For the sake of high robustness and easy implementation, we employ the local outlier factor (LOF) algorithm for fault diagnosis using $\mathbf{h}^n$. By conducting a representation dataset $\mathbb{H} = \{\mathbf{h}^1, \mathbf{h}^2, \cdots, \mathbf{h}^N\}$, the radius of the smallest hypersphere is calculated that contains the $k$ nearest neighbors. The local density is then computed, where a fault signal will have lower local density. One can refer to the work [26] for detailed steps.

## 4. Fault Data Analysis

Comprehensive experiments are performed to verify the effectiveness of the proposed fault diagnosis scheme. The electrical signals are collected from point machines that are installed and currently being used in operation lines. We have termed this current dataset Cu-3300, which contains three kinds of normal current classes and one fault class. The numbers of the signals in these normal classes are 474, 1043, and 1474, respectively, and the number of the signals in the fault class is 309. The exemplary comparison of normal signals and fault signals are shown in Figure 4. The number of the data points in the current signals differs from 92 to 300. The proposed locally connected autoencoder is trained using the Keras framework, where the CUDA acceleration technology is employed to accelerate the training process. The computational system environment is an Intel Core i9-9820X 3.30GHz processor running Ubuntu 18.04 LTS. The integrated development environment is Anaconda 5.3.1.

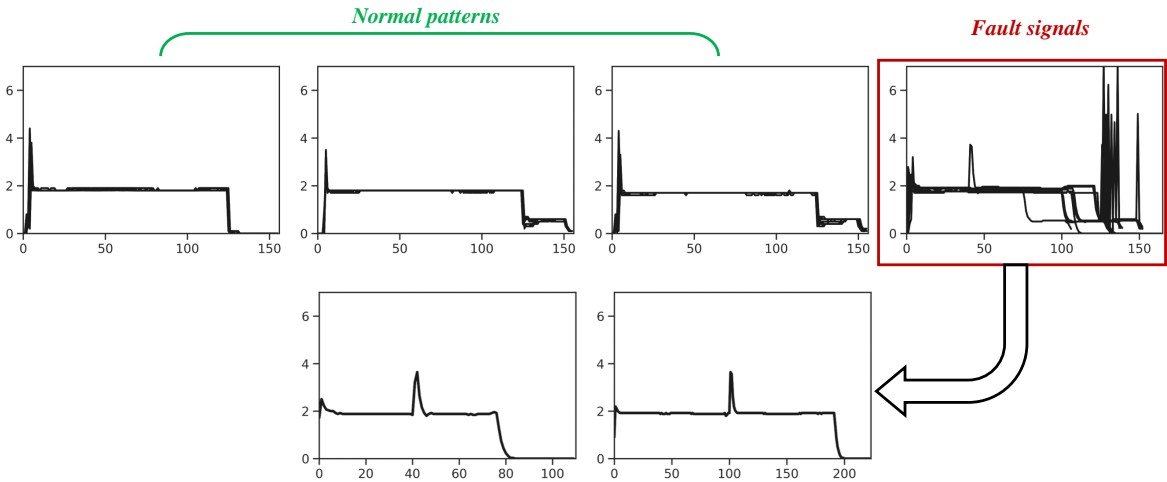

**Figure 4.** The exemplary comparison of normal signals and fault signals.

In traditional approaches, the features are manually designed and the PCA method is used for dimension reduction. In this paper, the fault diagnosis scheme termed "Featurebased" employing handcrafted features together with the kernel PCA method and the LOF algorithm is considered to be the baseline method. One can refer to our open source code for the detailed feature extraction process. Further, our proposed locally connected autoencoder scheme is also compared with other autoencoder-based methods:

1.　A stacked denoising autoencoder with L2 regularization (STDAE-150-50) [27], which adopts a two layer structure, with the first and the second layer containing, respectively, 150 and 50 units.
2.　A sparse denoising autoencoder with L1 and L2 regularizations (SPDAE-80) [28], in which the number of units is set to 80.

3. Two gated recurrent unit-based sequence to sequence autoencoders [29]. In the network structure, 100 units are employed in the encoder as well as the decoder. The autoencoders with 40 hidden units and 70 hidden units are named as GRU-40 and GRU-70, respectively.
4. The proposed fault diagnosis scheme without the weighting strategy.

*4.1. Feature Representations of Different Approaches*

To visualize the feature representations, similar to the work in [30], we map the representations in the multidimensional space to a two-dimensional plane using the kernel PCA method [31]. Kernel PCA transforms the data into the high-dimensional feature space and uses a Mercer kernel to reduce the dimension. Unless otherwise stated, the dimension reduction method used in this paper is the kernel PCA method.

Figure 5 shows the activations of the hidden representations of the baseline method and the contrast autoencoder-based methods for all 3300 sequences in $\mathbb{S}$. The normal classes are labeled as Class 1, Class 2 and Class 3, respectively. The fault class is flagged as Class -1. As can be seen in Figure 5, the principal components of Featurebased are more mixed compared to the autoencoder-based methods. This is because the hand-engineering feature extraction approach has difficulties handling the nonlinear data. Further, for the autoencoder-based methods, the current signals related to healthy point machines have a wide distribution, whereas the signals related to defective point machines have a more confined distribution, which may cause a decline of the effectiveness of the outlier detection. The latter is caused by violating a fundamental hypothesis in the local density-based method (such as the LOF algorithm) and the one class-based method (such as the ABOD algorithm [32]). The hypothesis is that in the feature space, the normal samples have higher density, whereas the abnormal samples have lower density [33].

The final feature representations of the proposed scheme without the weighting strategy are reported in Figure 6, where the number of the hidden units is set to 35. An overlapped window method [34] is adopted in the analysis, where the step size $S_{tr}$ changes from 20 to 40. Considering that the sequence length differs in Cu-3300, we preset a length parameter $L_{\text{pre}}$ to ensure the same length of sequences. To be more specific, if the length of $\mathbf{s}_n$ is less than $L_{\text{pre}}$, we pad the signals up to $L_{\text{pre}}$; when the length of $\mathbf{s}_n$ is greater than $L_{\text{pre}}$, the extra part is intercepted. Compared to the other autoencoder-based methods shown in Figure 5, the locally connected autoencoder performs slightly better, as the fault signal has a more dispersed distribution. The results demonstrate the great ability of the proposed method in automatically learning the useful features from the raw current signals. It also can be seen that the changes of $M_{\text{loc}}$ and $S_{\text{tr}}$ do not reduce the effectiveness of the proposed scheme, where the robustness is verified.

Figure 7 provides the feature representations of the proposed scheme, where $L = 10$ and $\alpha = 0.1$. Similar to Figure 6, the proposed scheme can achieve superior performance for the representation learning in the field of the point machine fault diagnosis. Note that using the weighting strategy, the densities of the feature representations of both the normal and fault signals are observed to be increased. The increase of the densities of both the normal and fault signals can be explained that the locally connected autoencoder with the weighting strategy incorporated can produce more effective representations. In Section 4.2, observe that the weighting strategy can effectively improve the detection accuracy, especially for those cases when the number of the neighbor points is relatively small. This is reasonable, as the increase of the densities of the feature representations caused by the weighting strategy is larger for the normal data than for the fault data. Also note that in this strategy, the weights of the informative signals are increased and the reason is similar to that of the adaboost algorithm [35].

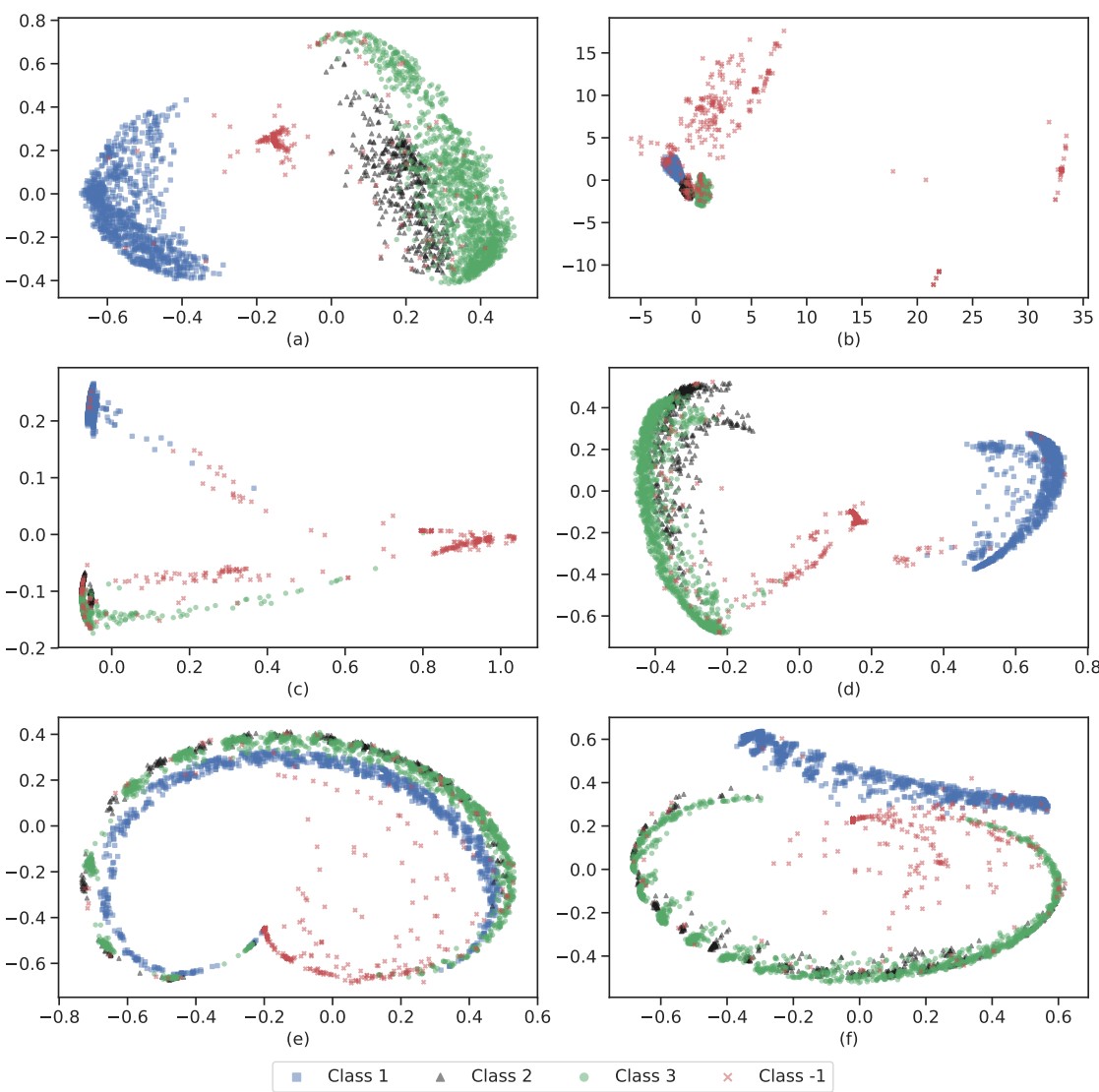

**Figure 5.** 2D projection of the final feature representations. (**a**) FeatureBased; (**b**) FeatureBased Method with PCA; (**c**) SPDAE-80; (**d**) STDAE-150-50; (**e**) GRU-40; (**f**) GRU-70.

### 4.2. Analysis Results of Fault Diagnosis

The results of the fault diagnosis are discussed in this subsection. As mentioned above, the LOF algorithm is employed for high robustness and easy implementation. A core parameter is the number of nearest neighbors in defining the outlier factor of the current signal, which is defined as $k$ in this paper.

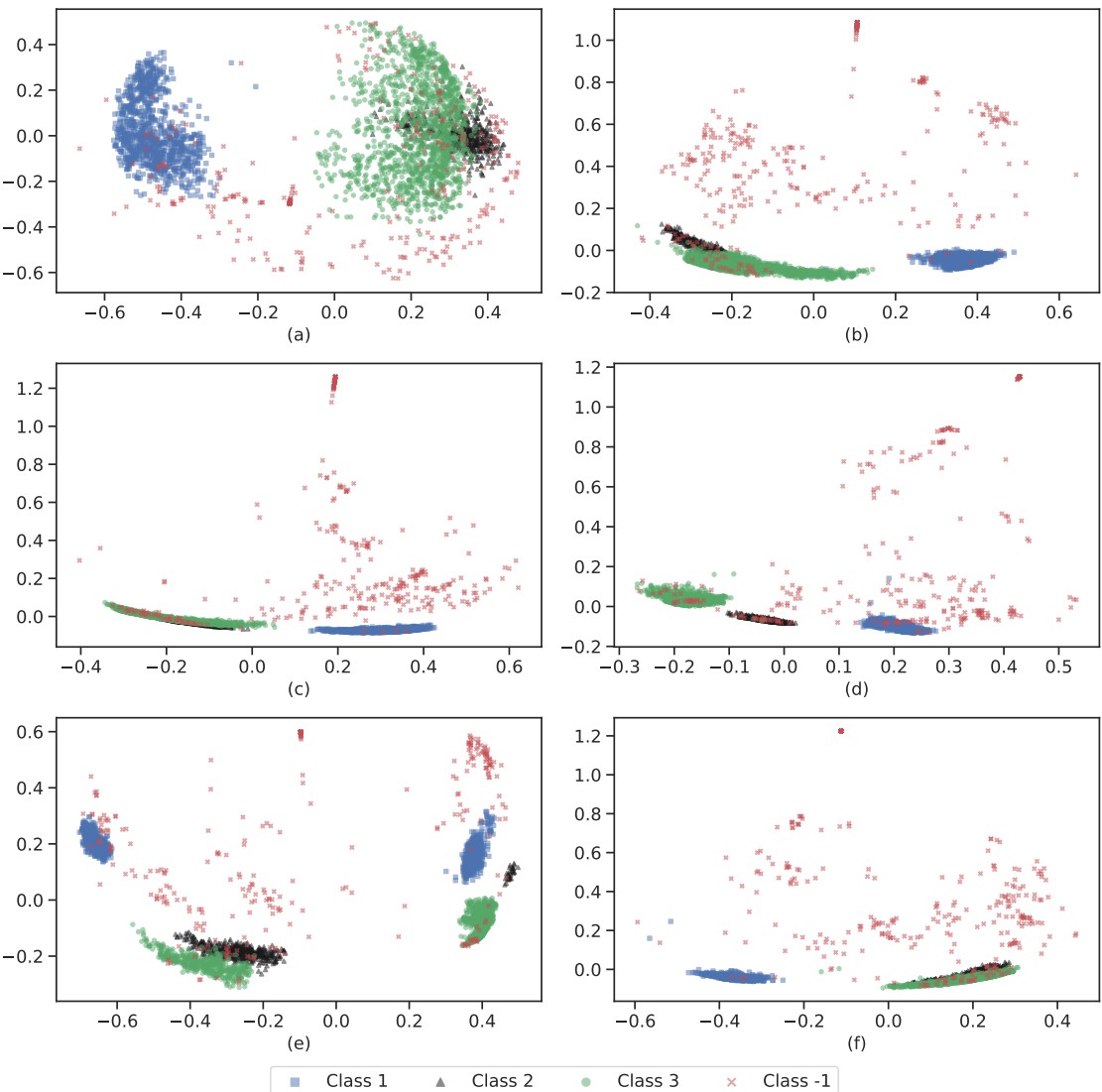

**Figure 6.** 2D projection of the final feature representations of the proposed scheme without the weighting strategy. (**a**) $M_{\text{loc}} = 20 = S_{\text{tr}} = 20$; (**b**) $M_{\text{loc}} = S_{\text{tr}} = 30$; (**c**) $M_{\text{loc}} = S_{\text{tr}} = 40$ ; (**d**) $M_{\text{loc}} = 40$, $S_{\text{tr}} = 10$; (**e**) $M_{\text{loc}} = 40$, $S_{\text{tr}} = 20$; (**f**) $M_{\text{loc}} = 40$, $S_{\text{tr}} = 30$.

The comparison of fault diagnosis of different schemes is reported in Figure 8, which shows the receiver operating characteristic (ROC) curves. The horizontal axis and the vertical axis of Figure 8 are the false positive rate and true positive rate, respectively. These curves are used to visualize the performance of the fault detection effect, as they measure the ability to produce good relative sample scores that serve to discriminate positive and negative samples [36]. To quantitatively measure the performance, the area under curve (AUC) is calculated. Bigger AUC values indicate better performances of the fault diagnosis scheme. For the analysis, $\alpha$ and $L$ are set as 0.1 and 10, respectively. The proposed scheme with $M_{\text{loc}} = 20$ and $S_{\text{tr}} = 20$ is named as Weight-I, whereas the proposed scheme with $M_{\text{loc}} = 30$ and $S_{\text{tr}} = 30$ is abbreviated as Weight-II. The corresponding schemes without the weighting strategy are termed as Average-I and Average-II, respectively. As can be seen in Figure 8, the parameter $k$ is set to 60, 80, 100, 140, and 190. The value of $k$ changes from 60 to 200. The optimal results of all the schemes are shown in Figure 8a. The figure shows that the locally connected autoencoder outperforms the other schemes as it has the highest AUC value of 0.981. The locally connected autoencoder without weighting strategy has the second best performance with

an AUC value of 0.979. The AUC scores of these schemes are listed in Table 1. Note that the scheme with the best performance is marked in bold in this table.

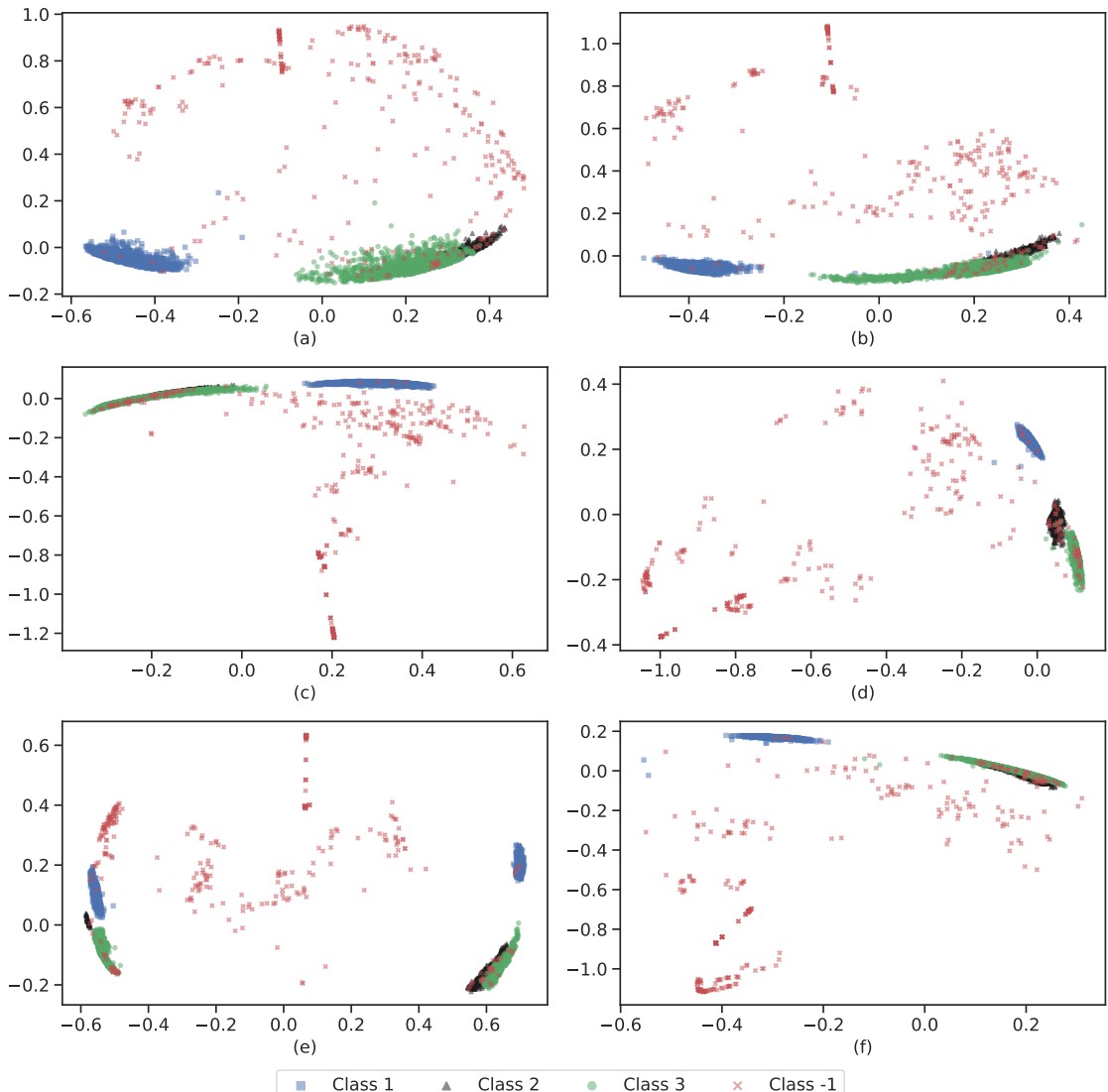

**Figure 7.** 2D projection of the final feature representations of the proposed scheme. (**a**) $M_{\text{loc}} = 20 = S_{\text{tr}} = 20$; (**b**) $M_{\text{loc}} = S_{\text{tr}} = 30$; (**c**) $M_{\text{loc}} = S_{\text{tr}} = 40$; (**d**) $M_{\text{loc}} = 40$, $S_{\text{tr}} = 10$; (**e**) $M_{\text{loc}} = 40$, $S_{\text{tr}} = 20$; (**f**) $M_{\text{loc}} = 40$, $S_{\text{tr}} = 30$.

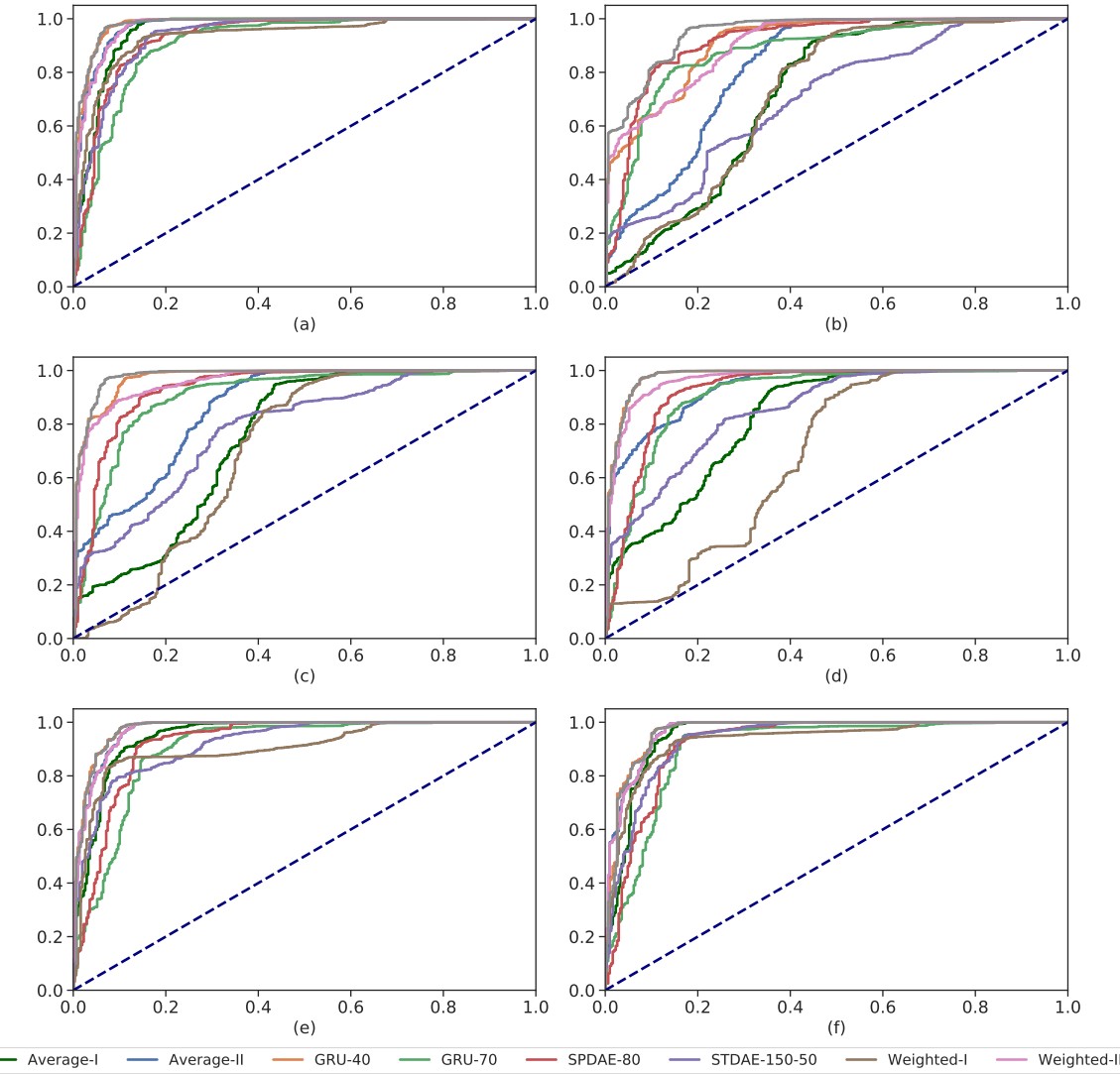

**Figure 8.** Receiver operating characteristic (ROC) curves and AUCs of different schemes. (**a**) optimal; (**b**) $k = 60$; (**c**) $k = 80$; (**d**) $k = 100$; (**e**) $k = 140$; (**f**) $k = 190$.

Figure 9 shows the impact of $\alpha$ and $L$ on the fault detection results of the locally connected autoencoder when the number of the hidden units is set equal to 35 and the values of $M_{\text{loc}}$ and $S_{\text{tr}}$ are both set to 30. From Figure 9, we observe that the locally connected autoencoder with the weighting strategy incorporated performs better than the scheme without the weight strategy. This is because this strategy enhances the densities of the feature representations of the normal signals, resulting in a stable performance of the fault diagnosis task. This fault analysis result also verifies the analysis of Figure 7. One can also find that, when $k$ is relativity small, the superiority of the proposed scheme is more notable, as the gaps of the AUC scores of the proposed scheme and the other schemes are larger. It is especially important as it is difficult to set appropriate values of $k$ in advance in actual fault diagnosis tasks. If a relativity small value of $k$ is selected, the proposed scheme retains a good performance while the other schemes suffer significant deterioration. When the value of $k$ increases to 120, the AUC scores are very close of the proposed scheme with or without the weighting strategy. Besides, both suffer a performance reduction when $k$ further increases. It also can be seen that the larger value of $\alpha$ causes the performance deterioration, which can be considered as "the overfitting phenomenon". The setting of $L$ also has an influence on the detection accuracy when $k$ is small. The delicate adjustment of $L$ is helpful to improve the performance. Our analysis result is consistent with the analysis in the original paper of LOF [26].

**Table 1.** Comparative results of different schemes in terms of AUC values.

| k | 60 | 80 | 100 | 140 | 190 |
|---|---|---|---|---|---|
| Featuresbased | 0.722 | 0.751 | 0.828 | 0.95 | 0.953 |
| SPDAE-80 | 0.708 | 0.783 | 0.854 | 0.923 | 0.936 |
| STDAE-150-50 | 0.719 | 0.703 | 0.689 | 0.905 | 0.935 |
| GRU-40 | 0.881 | 0.903 | 0.909 | 0.903 | 0.907 |
| GRU-70 | 0.916 | 0.930 | 0.926 | 0.924 | 0.923 |
| Average-I | 0.82 | 0.860 | 0.939 | 0.972 | **0.968** |
| Average-II | 0.912 | 0.973 | **0.979** | **0.976** | 0.97 |
| Weighted-I | 0.907 | 0.960 | 0.969 | 0.972 | **0.968** |
| Weighted-II | **0.949** | **0.981** | **0.979** | **0.976** | **0.968** |

Figure 10 reports the impact of the number of hidden units on the AUC scores when $\alpha$ and $L$ are fixed to 0.1 and 10, respectively, while $M_{\mathrm{loc}}$ and $S_{\mathrm{tr}}$ are set to 30. Then, for the number of hidden units equal to five, the AUC has a very small value, which can be explained by that a small number of hidden units implies relatively low dimensions of $\mathbf{W}$, $\mathbf{W}'$, $\mathbf{b}$ and $\mathbf{b}'$, which causes underfitting. When the number of units increases, the AUC scores stabilize at a relatively high value and so a high degree of accuracy is achieved. Further increasing the number of units results in decreasing AUC values, which is due to the problem of information redundancy. It shows that the excessive increase of the number of units negatively affects the fault detection. We conclude that through the performed fault analysis based on AUC scores the superiority of the locally connected autoencoder over the baseline method and other autoencoder-based methods has been verified.

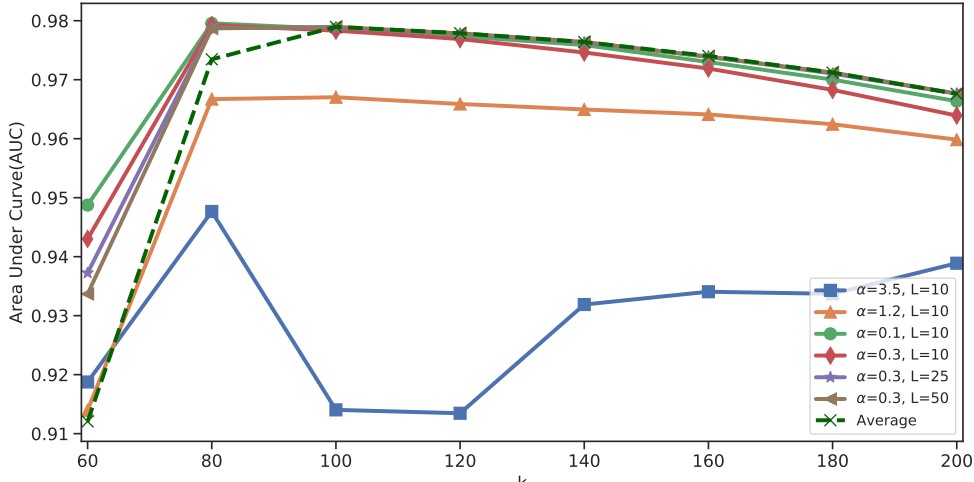

**Figure 9.** Performance comparison with different $\alpha$ and $L$.

In the process of the analysis, we have devoted much effort to test Cu-3300. Note that it is difficult to obtain accurate labels to indicate healthy or unhealthy states. Moreover, finding fault signals of all possible types of anomalous behaviors is even more difficult, as new types of faults may arise due to the dynamic characteristic. Thus, we propose that Cu-3300 may lack completeness. We have made the dataset Cu-3300 and the code file freely accessible online at https://github.com/MichaelYin1994/SignalRepresentationAnalysis.

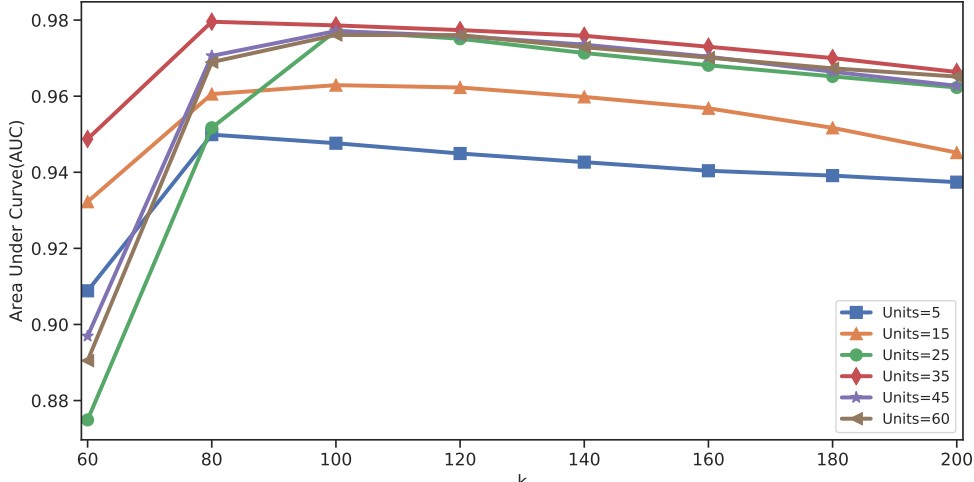

**Figure 10.** Performance comparison with different numbers of hidden units.

## 5. Conclusions

In this paper, a novel scheme using the locally connected autoencoder is proposed for intelligent fault diagnosis of railway point machines. To avoid the labor-intensive hand-engineering feature extraction, the proposed scheme is designed to automatically capture the informative features from raw electrical signals. Inspired by the word2vec method, in the locally connected autoencoder the input signals are segmented and blended to enhance the temporal characteristic. To further reduce the impact of the noninformative information in the electrical signals, a weighting strategy is proposed to increase the weights of useful and robust features, whereby the meaningful hidden representations of the raw signals can be obtained. Note that it is difficult to obtain labels for the training steps; in this paper, an unsupervised anomaly detection method (the LOF algorithm) is employed. We evaluate the prediction performance of the proposed scheme by comparing its prediction skills with respect to existing methods, such as the feature-based LOF algorithm, the stacked denoising autoencoder, the sparse denoising autoencoder, and the gated recurrent unit-based sequence to sequence autoencoder. Using the self-constructed fault dataset (Cu-3300), we demonstrate that the proposed scheme achieves superior performance with AUC = 0.981, compared to these methods. In the future work, we would investigate the optimization techniques to adaptively adjust the parameters of the locally connected autoencoder. Meanwhile, more studies are also planned to evaluate if the proposed scheme is applicable to other research fields.

**Author Contributions:** Conceptualization, T.T. and Z.L.; formal analysis, T.T. and Z.L.; methodology, Z.L. and Z.Y.; writing-original draft, Z.L.; validation, Z.L. and Z.Y.; resources, C.G.; supervision, C.G.

**Funding:** This research was funded by Beijing Postdoctoral Research Foundation (No. ZZ2019-117).

**Conflicts of Interest:** The authors declare no conflicts of interest.

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
