# Peer review of "Fault Diagnosis of Railway Point Machines Using the Locally Connected Autoencoder"

_applsci, doi:10.3390/app9235139_

Round 1

Reviewer 1 Report

The authors tackle the problem of fault diagnosis of railway point machines. For diagnosis, they utilised electric current signals registered during the operating phase of the point machine. Instead of conventional handcrafted feature extraction approach, they proposed a locally connected autoencoder to automatically capture high-order features. Moreover, their approach does not need the training step. The method was tested on an experimental set consisting of 3300 elements. The results confirm that their approach outperforms other methods.

The paper is well written and has sound motivation. The authors use modern tools which include automatic feature learning for the point machine diagnosis. Such an approach has not been reported so far in the literature.

I have only small remarks which should be considered before publication:

It would be interesting to show an exemplary comparison of raw healthy and faulty signals of the point machine. It would be easier to distinguish class separations in 4-6 if Class 2 is e.g. in yellow instead of orange which is close to the faulty case in red.

Author Response

Dear Reviewer,

We would like to express our sincere thanks to you for your time and efforts in reviewing our work. Thank you very much for your positive comments. They were gratifying. To improve the quality of the work, we have carefully revised the paper.

In the cover letter, we respond to your comments and concerns point-by-point. 

Reviewer 2 Report

Dear Authors,

Thank you for your paper and the nice work. Your paper is written clear and to the point, but still is slightly weak on a truly proper scientific formulation. Please find attached my nine page report consisting of many possible improvements that should be incorporated. My editing comments mainly refer to more scientific use of language, improving the figures and maybe extending the conclusion section slightly.

Further I mention here, that you only refer to equation (5) and (8) in the text of your paper, this means that actually you only need to number these two equations and all other equations do not need numbering.

KInd regards, reviewer 2

Author Response

Dear Reviewer,

First of all, we sincerely thank you individually for the painstaking review that you did.

Thank you very much for your positive comments. They were gratifying. We are also highly grateful for the constructive feedback, which helped to improve the quality of the work. Accordingly, we have revised the paper taking into consideration all the pertinent changes requested. During the modify process, your meaningful comments deepen our understanding of this work and help us to be more scientific.

Thank you very much, indeed!

In the cover letter, we respond to your comments and concerns. All the changes are highlighted in the revised version.
